# Biological Learning of Irreducible Representations of Commuting Transformations

**Alexander Genkin**[*]    **David Lipshutz**[†]    **Siavash Golkar**[†]

**Tiberiu Teşileanu**[†]    **Dmitri B. Chklovskii**[*,†]

[*]Neuroscience Institute, NYU Langone Medical School
[†]Center for Computational Neuroscience, Flatiron Institute

alexander.genkin@gmail.com
{dlipshutz,sgolkar,ttesileanu,dchklovskii}@flatironinstitute.org

## Abstract

A longstanding challenge in neuroscience is to understand neural mechanisms underlying the brain's remarkable ability to learn and detect transformations of objects due to motion. Translations and rotations of images can be viewed as orthogonal transformations in the space of pixel intensity vectors. Every orthogonal transformation can be decomposed into rotations within irreducible two-dimensional subspaces (or representations). For sets of commuting transformations, known as toroidal groups, Cohen and Welling proposed a mathematical framework for learning the irreducible representations. We explore the possibility that the brain also learns irreducible representations using a biologically plausible learning mechanism. The first is based on SVD of the anti-symmetrized outer product of the vectors representing consecutive images and is implemented by a single-layer neural network. The second is based on PCA of the difference between consecutive frames and is implemented in a two-layer network but with greater biological plausibility. Both networks learn image rotations (replicating Cohen and Welling's results) as well as translations. It would be interesting to search for the proposed networks in nascent connectomics and physiology datasets.

## 1 Introduction

The brains of humans and other animals identify objects, such as human faces, regardless of their location and orientation in the visual field. This requires learning identity-preserving transformations such as translations, scaling and rotations. Guiding locomotion requires identifying the direction and speed of the optic flow. Flies, to maintain stable flight, detect relative translational and rotational motion of their surroundings.

Whereas real-life transformations can be a mixture of translations and rotations, for simplicity, we consider learning one kind of transformation at a time. A sequence of pairs of images before and after transformations are streamed to an unsupervised algorithm. The algorithm learns these transformations and detects their magnitude, which corresponds to the speed times the time step.

Transformations such as rotations and translations of any magnitude can be decomposed into infinitesimal transformation operators (called generators) and magnitudes. For these so called Lie groups, generators can be learned from data and magnitudes of transformation determined for each pair of images, as shown by Rao and Ruderman [14].

36th Conference on Neural Information Processing Systems (NeurIPS 2022).

Many transformations can be represented as rotations in the high-dimensional space of vectorized images. Such rotations can be decomposed into a set of commuting 2D rotations, each characterized by a generator and a magnitude, and comprise toroidal groups. Learning toroidal groups reduces to finding an "irreducible representation"—a problem solved by Bethge et al. [3] (for fixed magnitudes) and Cohen and Welling [5] (for varying magnitudes).

An important problem in neuroscience is to develop biologically plausible algorithms and neural circuits that learn to detect transformations and their magnitudes. By biological plausibility we understand the following requirements [4]: (i) the algorithm operates in the online setting where pairs of images are streamed sequentially while the transformations are learned and the magnitudes are determined in real time; (ii) previously seen data cannot be stored by the algorithm other than a highly compressed representation in the synaptic weights; (iii) synaptic weight update (learning) rules are local, meaning that the weight update at a particular synapse depends only on the activity of the two neurons it connects.

The generator approach of [14] presents difficulties for biologically plausible neural circuit implementations. Specifically, detecting transformation magnitude requires multiplying all possible pairs of pixels taken from the images before and after the transformation weighted by the generator. The difficulty was overcome in two biologically plausible neural circuits proposed in [1].

In this paper we propose two algorithms for learning irreducible representations of toroidal groups that exhibit greater biological plausibility than prior work. We construct biologically plausible implementations that build upon the work from Cohen and Welling [5], which did not address biological plausibility. In contrast to the generator-based approach [14; 1], using irreducible representations eliminates the need for the multiplication of inputs. The neural circuits we propose can be searched for in the nascent connectomics and physiology datasets.

The first proposal is based on computing an average of anti-symmetrized outer products of vectorized images (bivectors, as in [3]) followed by a singular value decomposition (SVD) to yield irreducible representations. To perform SVD online, we adapt an algorithm for the SVD of cross-covariance matrix. When only the top irreducible component is desired, we show how to implement this algorithm using a biologically plausible model of a single neuron. For multiple irreducible representations the algorithm relies on deflation. The neural network implementation of the deflation step violates requirement (iii) of biological plausibility above, which motivates another approach.

Our second proposal is based on the PCA of time differences of vectorized images. The corresponding neural network relies solely on local learning rules, which makes it fully biologically plausible by our definition. The resulting two-layer network decomposes the transformations into their irreducible 2D rotations in the first layer, and outputs the angle of each 2D rotation in the second layer.

## 2 Setup

Consider a stream of consecutive video frames (Fig. 1) vectorized into $d$-dimensional pixel-intensity vectors $\{\mathbf{x}_t\}$ satisfying linear relations

$$\mathbf{x}_t = \mathbf{A}_t \mathbf{x}_{t-1}. \tag{1}$$

We focus here on image transformations that maintain the norm of $\mathbf{x}_t$ approximately constant. These include translations, rotations, and more generally, local deformations, which shuffle pixels around without affecting their values significantly. This assumption implies that the transformation matrix $\mathbf{A}_t$ is orthogonal. In this case, we can decompose $\mathbf{A}_t$ into independent rotations within orthogonal 2D subspaces, as follows

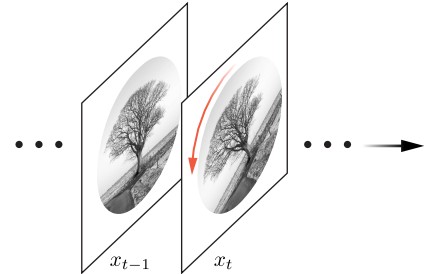

Figure 1: Consecutive video frames with rotated images are streamed to the learning algorithms.

$$\mathbf{A}_t = \mathbf{Q}_t \mathbf{\Gamma}_t \mathbf{Q}_t^\top, \quad \mathbf{\Gamma}_t = \text{blockdiag}(\mathbf{R}(\theta_t^1), \ldots, \mathbf{R}(\theta_t^{d/2})), \quad \mathbf{R}(\theta) := \begin{pmatrix} \cos\theta & -\sin\theta \\ \sin\theta & \cos\theta \end{pmatrix}. \tag{2}$$

Here $\mathbf{Q}_t$ is a $d \times d$ orthogonal matrix, $\mathbf{\Gamma}_t$ is a $d \times d$ block-diagonal matrix whose blocks are $2 \times 2$ rotation matrices indexed by $\theta_t^i$, $i = 1, \ldots, d/2$, and we have assumed, for simplicity, that $d$ is even.

Pairs of consecutive column vectors $(\mathbf{q}_{t,2i-1}, \mathbf{q}_{t,2i})$ of $\mathbf{Q}_t$ define orthogonal 2D subspaces and $\theta_t^i$ represents the angle of rotation within that 2D subspace. As desired, this canonically separates the transformation identity (the 2D subspace) from the velocity (the angle of rotation).

Following Cohen and Welling [5], we consider the problem of learning a representation for the $d/2$-dimensional commutative subgroup of rotations of the form $\mathbf{A}_t = \mathbf{Q}\boldsymbol{\Gamma}_t\mathbf{Q}^\top$, where $\mathbf{Q}$ is a *fixed* orthogonal matrix and $\boldsymbol{\Gamma}_t$ is any block-diagonal matrix of the form in equation (2). The subgroup is commonly referred to as a *toroidal subgroup* because it is parameterized by the $d/2$-dimensional torus $\{(\theta^1, \ldots, \theta^{d/2}) : \theta^i \in [0, 2\pi)\}$. With this simplification, the $d/2$ mutually orthogonal 2D subspaces constitute an *irreducible representation* of the subgroup [5].

The focus of this work is to derive a neurally plausible algorithm for (A) learning the irreducible representation from a stream of pairs of vectorized video frames $\{(\mathbf{x}_{t-1}, \mathbf{x}_t)\}$, and (B) outputting the angle of rotation $\theta_t^i$ within the top $k$ 2D subspaces, for some $1 \le k \le d/2$.

## 3 Learning the irreducible representation from data

We consider learning the irreducible representations from data in the offline setting using two approaches. The first approach is based on an SVD of the average bivector between consecutive frames. The second approach is based on PCA of the covariance of the difference between consecutive frames. The analysis here will be the starting point for the derivations of the online algorithms.

We assume the sequence $\{\mathbf{x}_t\}$ is centered and whitened so that $\langle \mathbf{x}_t\mathbf{x}_t^\top\rangle_t = \mathbf{I}_d$. Whitening of sensory inputs is hypothesized to be a function of early sensory systems [2] so we believe it is reasonable to assume that the circuit receives pre-whitened inputs. Furthermore, a number of biologically plausible algorithms for input whitening have been proposed and experimentally observed in some cases [6; 11; 16].

In addition, we assume that $\mathbf{A}_t$ is independent of $\mathbf{x}_t$ [14; 3], which holds approximately in many situations. For instance, most objects can move either left, right or up, down, even though the exact probabilities of each kind of transformation might depend weakly on object identity.

### 3.1 Learning on SVD of bivectors of consecutive frames

Given a pair of consecutive frames $(\mathbf{x}_{t-1}, \mathbf{x}_t)$, define the *bivector*
$$\mathbf{B}_t := \mathbf{x}_t\mathbf{x}_{t-1}^\top - \mathbf{x}_{t-1}\mathbf{x}_t^\top = \mathbf{A}_t\mathbf{x}_{t-1}\mathbf{x}_{t-1}^\top - \mathbf{x}_{t-1}\mathbf{x}_{t-1}^\top\mathbf{A}_t^\top,\tag{3}$$
where we have used the relation $\mathbf{x}_t = \mathbf{A}_t\mathbf{x}_{t-1}$. The bivector is an anti-symmetric matrix, which is also commonly referred to as the 'wedge product' or 'exterior product'. Under our assumptions that $\mathbf{x}_t$ and $\mathbf{A}_t$ are independent and $\{\mathbf{x}_t\}$ is whitened (i.e., $\langle \mathbf{x}_t\mathbf{x}_t^\top\rangle_t = \mathbf{I}_d$), we have
$$\mathbf{B} := \langle \mathbf{B}_t\rangle_t = \langle \mathbf{A}_t - \mathbf{A}_t^\top\rangle_t = \mathbf{Q}\mathbf{G}\mathbf{Q}^\top,\tag{4}$$
where
$$\mathbf{G} := \langle \boldsymbol{\Gamma}_t - \boldsymbol{\Gamma}_t^\top\rangle_t = \mathrm{blockdiag}(\mathbf{G}_1, \ldots, \mathbf{G}_k), \quad \mathbf{G}_i := \begin{pmatrix} 0 & -g_i \\ g_i & 0 \end{pmatrix}, \quad g_i := 2\langle \sin\theta_t^i\rangle_t.$$

Note that $g_i$ can be zero if the distribution of $\theta_t^i$ is symmetric about zero. Therefore, to use this approach, we assume that $g_i \ne 0$ for $i = 1, \ldots, k$.

To learn the column vectors of $\mathbf{Q} = [\mathbf{q}_1, \ldots, \mathbf{q}_d]$, it will be convenient to write the SVD of $\mathbf{B}$ in terms of $\mathbf{Q}$. To this end, we assume, without loss of generality, that $g_1 \ge \cdots \ge g_{d/2} \ge 0$. Then we have the following lemma, whose proof is deferred to Supplement A.

**Lemma 1.** *The SVD of $\mathbf{B}$ can be written as $\mathbf{B} = \mathbf{U}\boldsymbol{\Sigma}\mathbf{V}^\top$, where*
$$\mathbf{U} := \mathbf{Q}, \qquad \boldsymbol{\Sigma} := \mathrm{diag}(g_1, g_1, \ldots, g_{d/2}, g_{d/2}), \qquad \mathbf{V} := [\mathbf{v}_1, \ldots, \mathbf{v}_d],$$
*and $(\mathbf{v}_{2i-1}, \mathbf{v}_{2i}) = (\mathbf{q}_{2i}, -\mathbf{q}_{2i-1})$ for $i = 1, \ldots, d/2$.*

In other words, each pair of left- and right-singular vectors of $\mathbf{B}$, $(\mathbf{u}_i, \mathbf{v}_i)$, spans one of the 2D mutually orthogonal subspaces. Therefore, to learn the irreducible representation, it suffices to learn pairs of left- and right-singular vectors of $\mathbf{B}$. Moreover, the first pair of left- and right-singular vectors spans the same subspace as the second pair of left- and right-singular vectors, and same with the third and fourth pairs, and so on. Therefore it is sufficient to learn only a subset such as all odd-numbered pairs.

## 3.2   Learning on PCA of differences between consecutive frames

We now present an alternative method for learning the column vectors of $\mathbf{Q}$, by performing PCA on the covariance of the discrete time derivatives defined by $\dot{\mathbf{x}}_t := \mathbf{x}_t - \mathbf{x}_{t-1}$. Under our assumptions that $\mathbf{A}_t$ is independent of $\mathbf{x}_t$ and the vectors $\mathbf{x}_t$ are whitened, we have

$$\langle \dot{\mathbf{x}}_t \dot{\mathbf{x}}_t^\top \rangle_t = 2\mathbf{I}_d - \langle \mathbf{A}_t + \mathbf{A}_t^\top \rangle_t = \mathbf{Q}(2\mathbf{I}_d - \langle \mathbf{\Gamma}_t + \mathbf{\Gamma}_t^\top \rangle_t)\mathbf{Q}^\top = \mathbf{Q}\hat{\mathbf{\Gamma}}\mathbf{Q}^\top, \tag{5}$$

where $\hat{\mathbf{\Gamma}} = \mathrm{diag}(\gamma_1, \gamma_1, \dots, \gamma_k, \gamma_k)$ and $\gamma_i := 2 - 2\langle \cos \theta_t^i \rangle_t$. Importantly, the column vectors of $\mathbf{Q}$ are eigenvectors of $\langle \dot{\mathbf{x}}_t \dot{\mathbf{x}}_t^\top \rangle_t$ with associated eigenvalues $\gamma_i$ and, assuming that $\gamma_1 > \cdots > \gamma_{d/2}$, each $\gamma_i$ corresponds to a unique 2D subspace. Therefore the transformations with the largest angles can be found by performing PCA on $\langle \dot{\mathbf{x}}_t \dot{\mathbf{x}}_t^\top \rangle_t$. Note that under this approach, $\gamma_i = 0$ if and only if $\theta_t^i$ is identically zero.

It is worth noting that, under the whitening assumption on $\mathbf{x}_t$, performing PCA on the discrete time sums $\mathbf{x}_t + \mathbf{x}_{t-1}$ learns features that are 'temporally slow' [10], which is hypothesized to be useful for learning objects [17]. From this perspective, learning transformations can be viewed as a complement of learning objects.

# 4   Biological algorithms based on SVD of the average bivector

To compute the SVD of $\mathbf{B}$, consider the following optimization problem:

$$\max_{\mathbf{U}, \mathbf{V}} \mathrm{Tr}(\mathbf{U}^\top \mathbf{B} \mathbf{V}) \qquad \text{s.t.} \qquad \mathbf{U}^\top \mathbf{U} = \mathbf{V}^\top \mathbf{V} = \mathbf{I}_d, \tag{6}$$

whose solution is the pair of matrices of left- and right-singular vectors of $\mathbf{B}$. To solve this online we adapt an algorithm for SVD of cross-covariance matrix [7].

## 4.1   Finding the subspace with largest rotation velocity

Start with an optimization problem for finding the top pair of singular vectors:

$$\max_{\mathbf{u}, \mathbf{v}} \mathbf{u}^\top \mathbf{B} \mathbf{v} \qquad \text{s.t.} \qquad \|\mathbf{u}\| = \|\mathbf{v}\| = 1. \tag{7}$$

In the online setting optimization is performed by projected stochastic gradient descent:

$$\mathbf{u} \leftarrow \frac{\mathbf{u} + \eta_t \mathbf{B}_t \mathbf{v}}{\|\mathbf{u} + \eta_t \mathbf{B}_t \mathbf{v}\|}, \qquad\qquad \mathbf{v} \leftarrow \frac{\mathbf{v} - \eta_t \mathbf{B}_t \mathbf{u}}{\|\mathbf{v} - \eta_t \mathbf{B}_t \mathbf{u}\|}, \tag{8}$$

where $\eta_t > 0$ is the learning rate, $\mathbf{B}_t$ is used as the current approximation of $\mathbf{B}$, and we have used the anti-symmetric property $\mathbf{B}_t^\top = -\mathbf{B}_t$.

Normalization steps are hard to map onto a biological circuit, so following the idea from Oja's algorithm [13]: assuming small enough learning rates, update equations can be expanded using a Taylor series in $\eta_t$. In this case, the iteration takes the form:

$$\mathbf{u} \leftarrow \mathbf{u} + \eta_t \big(\mathbf{B}_t \mathbf{v} - (\mathbf{u}^\top \mathbf{B}_t \mathbf{v})\mathbf{u}\big), \tag{9a}$$

$$\mathbf{v} \leftarrow \mathbf{v} + \eta_t \big(-\mathbf{B}_t \mathbf{u} + (\mathbf{v}^\top \mathbf{B}_t \mathbf{u})\mathbf{v}\big). \tag{9b}$$

A convergence proof for this algorithm is given in Supplement B.

Since the matrix $\mathbf{B}$ is anti-symmetric, the singular vectors resulting from this optimization, $\mathbf{u}$ and $\mathbf{v}$, are orthogonal, and define the 2D subspace with the largest average sine of the angle of rotation. Now each input vector $\mathbf{x}_t$ can be projected on the 2D subspace, and we denote the projected coordinates as $a_t := \mathbf{u}^\top \mathbf{x}_t, b_t := \mathbf{v}^\top \mathbf{x}_t$. Using the definition of $\mathbf{B}_t$ from Eq. (3) and substituting $a_t, b_t$ into the objective yields:

$$\mathbf{u}^\top \mathbf{B} \mathbf{v} = \mathbf{u}^\top \langle \mathbf{B}_t \rangle_t \mathbf{v} = \langle a_t b_{t-1} - a_{t-1} b_t \rangle_t = \langle y_t \rangle_t, \tag{10}$$

where $y_t := a_t b_{t-1} - a_{t-1} b_t$. This suggests two linear neurons with common input and weights $\mathbf{u}$ and $\mathbf{v}$ generating output streams $a_t$ and $b_t$, followed by a circuit known as a Hassenstein-Reichardt detector [15]; see Fig. 2(a). Now the angle of rotation $\theta_t$ can be recovered:

$$\theta_t = \arcsin\left(\frac{y_t}{\sqrt{a_t^2 + b_t^2}\sqrt{a_{t-1}^2 + b_{t-1}^2}}\right).$$

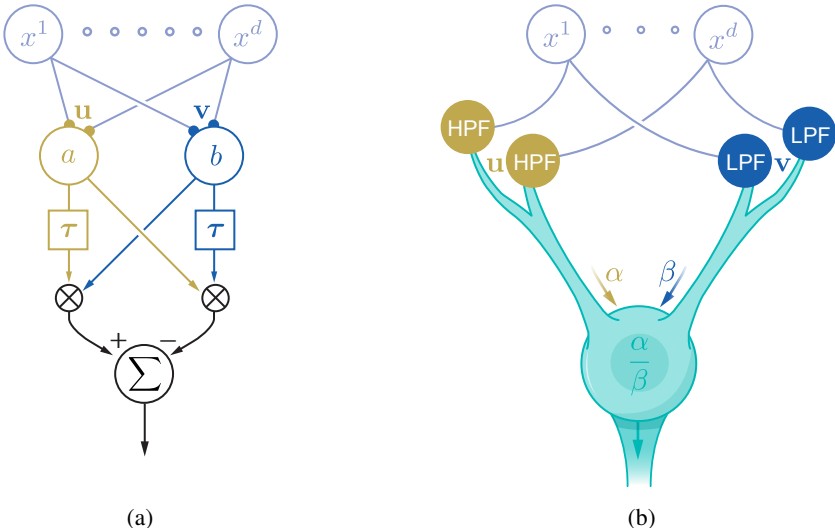

(a)                                        (b)

Figure 2: Biologically plausible implementations of the SVD algorithm. (a) The Hassenstein-Reichardt circuit calculates bivectors of consecutive inputs. Neuron $a$ ($b$) projects input vectors onto the synaptic weight vector $\mathbf{u}$ ($\mathbf{v}$). Then the delayed and non-delayed signals from $a$ and $b$ are multiplied and the products are ultimately subtracted, Eq. (10). (b) A neuron learning the subspace with largest rotation velocity. Left (right) dendritic branches receive synapses projecting the input onto the vector $\mathbf{u}$ ($\mathbf{v}$) and perform high- (low-) pass filtering (HPF and LPF in the figure) resulting in dendritic current $\alpha$ ($\beta$). The two dendritic currents are divided in the soma.

## 4.2 Full dimension: finding multiple subspaces

Subsequent singular vector pairs can be found by deflation. Recall the identities for anti-symmetric SVD: $\mathbf{u}_{2i-1} = \mathbf{v}_{2i}, \mathbf{u}_{2i} = -\mathbf{v}_{2i-1}, \sigma_{2i-1} = \sigma_{2i} = g_i$ as in Lemma 1. We can thus rewrite SVD using odd indexes only:

$$\mathbf{B} = \sum_i \sigma_i \mathbf{u}_i \mathbf{v}_i^\top = \sum_{j:\text{odd}} (\mathbf{u}_j \mathbf{v}_j^\top - \mathbf{v}_j \mathbf{u}_j^\top) g_{(j+1)/2} = \sum_{j:\text{odd}} (\mathbf{u}_j \mathbf{u}_j^\top + \mathbf{v}_j \mathbf{v}_j^\top) \mathbf{B} \,. \qquad (11)$$

Then $\mathbf{u}_i, \mathbf{v}_i$ for $i$ odd, which is all we need, can be found as solution of an optimization problem similar to Eq. (7), but for the "deflated" matrix:

$$\max_{\mathbf{u},\mathbf{v}} \mathbf{u}^\top \Big( \mathbf{I} - \sum_{j:\text{odd}, j<i} (\mathbf{u}_j \mathbf{u}_j^\top + \mathbf{v}_j \mathbf{v}_j^\top) \Big) \mathbf{B} \mathbf{v} \qquad \text{s.t.} \quad \|\mathbf{u}\| = \|\mathbf{v}\| = 1 \,. \qquad (12)$$

---

**Algorithm 1:** The SVD algorithm with deflation

**for** $i = 1$ to $k$ **do**
    $\mathbf{u}_i, \mathbf{v}_i \leftarrow$ random, $\|\mathbf{u}_i\| = \|\mathbf{v}_i\| = 1$ {Initialize}
**end for**
**for** $t = 1, 2, \dots$ **do**
    $\mathbf{B}_t \leftarrow \mathbf{x}_t \mathbf{x}_{t-1}^\top - \mathbf{x}_{t-1} \mathbf{x}_t^\top$
    $\mathbf{D} \leftarrow 0$ {current deflation estimate}
    **for** $i = 1$ to $k$ **do**
        $\mathbf{u}_i \leftarrow \mathbf{u}_i + \eta_t [\mathbf{B}_t \mathbf{v}_i - \mathbf{u}_i \mathbf{u}_i^\top \mathbf{B}_t \mathbf{v}_i - \mathbf{D} \mathbf{B}_t \mathbf{v}_i]$
        $\mathbf{v}_i \leftarrow \mathbf{v}_i + \eta_t [-\mathbf{B}_t \mathbf{u}_i + \mathbf{v}_i \mathbf{v}_i^\top \mathbf{B}_t \mathbf{u}_i + \mathbf{D} \mathbf{B}_t \mathbf{u}_i]$
        $\mathbf{D} \leftarrow \mathbf{D} + \mathbf{u}_i \mathbf{u}_i^\top + \mathbf{v}_i \mathbf{v}_i^\top$ {update deflation estimate}
        $a_t^i \leftarrow \mathbf{u}_i^\top \mathbf{x}_t, \quad b_t^i \leftarrow \mathbf{v}_i^\top \mathbf{x}_t$
        $\theta_t^i \leftarrow \arcsin\left( \dfrac{a_t^i b_{t-1}^i - a_{t-1}^i b_t^i}{\sqrt{a_t^{i\,2} + b_t^{i\,2}} \sqrt{a_{t-1}^{i\,2} + b_{t-1}^{i\,2}}} \right)$
    **end for**
**end for**

---

A straightforward deflation algorithm would find singular vector pairs one by one, corresponding to decreasing singular values. The goal however is to find all of them in parallel, in an online and neurally plausible manner. On every time step, the algorithm will perform one step of the iterations from Eqns. (9a) and (9b) in a loop on sequentially deflated matrices, using current estimates of singular vectors. Using again $\mathbf{B}_t$ as the current approximation of $\mathbf{B}$, this yields Algorithm 1. Proof of convergence can be found in the Supplement B.

### 4.3 Biological SVD

The circuit in Fig. 2(a) above is not realizable inside a neuron because of the crossing pathways. It also cannot be realized with two neurons, as the original Hassenstein-Reichardt detector, because then the synaptic weight updates in one neuron would require information only available in the other neuron, violating the principle of locality. To overcome these difficulties we propose a modification of the bivector expression:

$$\tilde{\mathbf{B}}_t = (\mathbf{x}_t - \mathbf{x}_{t-1})(\mathbf{x}_t + \mathbf{x}_{t-1})^\top . \tag{13}$$

This is similar to what was used in [1], but with important difference: the expectation under our assumptions is the same as in Eq. (4) (though individual matrices are no longer anti-symmetric). Projected on the 2D subspace of interest are now the difference and sum of vectors: $\alpha_t := \mathbf{u}^\top(\mathbf{x}_t - \mathbf{x}_{t-1})$ and $\beta_t := \mathbf{v}^\top(\mathbf{x}_t + \mathbf{x}_{t-1})$. Substituting into the objective, we can rewrite: $\mathbf{u}^\top \tilde{\mathbf{B}}_t \mathbf{v} = \alpha_t \beta_t$.

This suggests a neuron with two dendritic branches both connecting to the same set of inputs, Fig. 2(b). The first has synapses with weights $\mathbf{u}$ that also calculate temporal differences (perform low-pass filtering); the second one has synapses with weights $\mathbf{v}$ that also perform high-pass filtering. The neuron performs division in the soma and outputs $y_t = \alpha_t/\beta_t = \tan\theta_t$, from which the angle can be computed. We assume the activity is suppressed when $\beta_t$ is too small for reliable calculation of the ratio. That means the neuron has a "blind spot" when $(\mathbf{x}_t + \mathbf{x}_{t-1})$ is close to orthogonal to $\mathbf{v}$.

The neural dynamics are described by the equations:

$$\begin{aligned}
\alpha_t &\leftarrow \mathbf{u}^\top(\mathbf{x}_t - \mathbf{x}_{t-1}) \\
\beta_t &\leftarrow \mathbf{v}^\top(\mathbf{x}_t + \mathbf{x}_{t-1}) \\
y_t &\leftarrow \alpha_t/\beta_t \,,
\end{aligned} \tag{14}$$

and synaptic plasticity rules are derived using a Taylor approximation in the learning rate, as in Eqs. (9a)–(9b):

$$\begin{aligned}
\mathbf{u} &\leftarrow \mathbf{u} + \eta_t\big(\beta_t(\mathbf{x}_t - \mathbf{x}_{t-1}) - \alpha_t\beta_t\mathbf{u}\big) , \\
\mathbf{v} &\leftarrow \mathbf{v} + \eta_t\big(\alpha_t(\mathbf{x}_t + \mathbf{x}_{t-1}) + \alpha_t\beta_t\mathbf{v}\big) .
\end{aligned} \tag{15}$$

## 5 Biological algorithm based on PCA of time differences

In this section, we derive a two-layer network that outputs the angles $\theta^1, \ldots, \theta^k$. The first layer of our network projects the inputs $x^1, \ldots, x^d$ onto the top $k$ 2D subspaces defined by the pairs of column vectors $\mathrm{span}(\mathbf{q}_1, \mathbf{q}_2), \ldots, \mathrm{span}(\mathbf{q}_{2k-1}, \mathbf{q}_{2k})$, and the second layer computes the angles of rotation within each 2D subspace.

### 5.1 Derivation of the first layer

Recall that for each $i = 1, \ldots, k$, the vectors $(\mathbf{q}_{2i-1}, \mathbf{q}_{2i})$ are eigenvectors of the covariance matrix $\langle \dot{\mathbf{x}}_t \dot{\mathbf{x}}_t^\top \rangle_t$ associated with eigenvector $\gamma_i$. To project the inputs $\mathbf{x}_t$ onto the 2D subspaces, we adopt the similarity matching objective introduced in [12]. We assume the first layer of the network has $2k$ neurons whose activities at time $t$ are encoded in the vector $\mathbf{y}_t \in \mathbb{R}^{2k}$. Define the data matrices $\mathbf{X} := [\mathbf{x}_1, \ldots, \mathbf{x}_T]$, $\mathbf{Y} := [\mathbf{y}_1, \ldots, \mathbf{y}_T]$, $\dot{\mathbf{X}} := [\dot{\mathbf{x}}_1, \ldots, \dot{\mathbf{x}}_T]$ and $\dot{\mathbf{Y}} := [\dot{\mathbf{y}}_1, \ldots, \dot{\mathbf{y}}_T]$, where recall that the dot represents a discrete time derivative, e.g., $\dot{\mathbf{x}}_t = \mathbf{x}_t - \mathbf{x}_{t-1}$. Let $\dot{\mathbf{X}} = \mathbf{Q}\boldsymbol{\Sigma}\mathbf{P}^\top$ denote the SVD of the data matrix. We start with the PCA objective function [12]

$$\min_{\mathbf{Y}} -2\,\mathrm{Tr}(\dot{\mathbf{X}}^\top\dot{\mathbf{X}}\dot{\mathbf{Y}}^\top\dot{\mathbf{Y}}) + \mathrm{Tr}(\dot{\mathbf{Y}}^\top\boldsymbol{\Lambda}^{-1}\dot{\mathbf{Y}}\dot{\mathbf{Y}}^\top\boldsymbol{\Lambda}^{-1}\dot{\mathbf{Y}}), \tag{16}$$

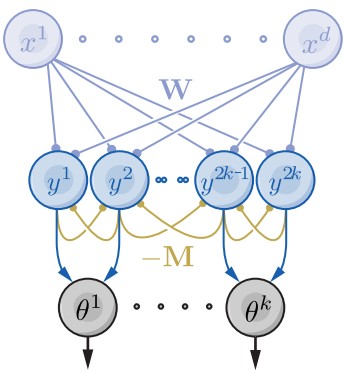

Figure 3: A two-layer network for learning transformations using the PCA algorithm. In the first layer, inputs are projected onto multiple subspaces. In the second layer, the rotation angles $\theta_t^1, \ldots, \theta_t^k$ are computed according to Algorithm 2.

where $\boldsymbol{\Lambda} = \mathrm{diag}(\lambda_1, \lambda_1, \ldots, \lambda_k, \lambda_k)$ is any fixed diagonal matrix with $\lambda_1 > \cdots > \lambda_k$. Suppose that the diagonal elements $\gamma_i$ of the matrix $\hat{\boldsymbol{\Gamma}}$ (see Eq. (5)) obey $\gamma_1 > \gamma_2 > \cdots > \gamma_k$. Then the optimal solutions to (16) are given by

$$\mathbf{Y}_{\mathrm{opt}} = \boldsymbol{\Lambda}\mathbf{R}\mathbf{Q}^\top\mathbf{X}, \tag{17}$$

where $\mathbf{R}$ is any block-diagonal matrix with $2 \times 2$ blocks that are orthogonal matrices, see [12, Lemma 1]. In particular, the optimal solution is the projection of the data matrix $\mathbf{X}$ onto the 2D subspaces defined by the pairs of column vectors of $\mathbf{Q}$.

Directly optimizing (16) will not result in an online algorithm due to the sample covariance terms $\frac{1}{T}\dot{\mathbf{X}}\dot{\mathbf{Y}}^\top$ and $\frac{1}{T}\dot{\mathbf{Y}}\dot{\mathbf{Y}}^\top$. Therefore, to obtain an online algorithm, we encode the sample covariances in the matrices $\mathbf{W} \in \mathbb{R}^{2k \times d}$ and $\mathbf{M} \in \mathbb{R}^{2k \times 2k}$ by substituting in with the following Legendre transforms

$$\frac{1}{T^2}\mathrm{Tr}(\dot{\mathbf{X}}^\top\dot{\mathbf{X}}\dot{\mathbf{Y}}^\top\dot{\mathbf{Y}}) = \min_{\mathbf{W} \in \mathbb{R}^{2k \times d}} \left\{ \frac{2}{T}\mathrm{Tr}(\dot{\mathbf{Y}}^\top\mathbf{W}\dot{\mathbf{X}}) - \mathrm{Tr}(\mathbf{W}\mathbf{W}^\top) \right\}$$

$$\frac{1}{T^2}\mathrm{Tr}(\dot{\mathbf{Y}}^\top\boldsymbol{\Lambda}^{-1}\dot{\mathbf{Y}}\dot{\mathbf{Y}}^\top\boldsymbol{\Lambda}^{-1}\dot{\mathbf{Y}}) = \min_{\mathbf{M} \in \mathbb{R}^{2k \times 2k}} \left\{ \frac{2}{T}\mathrm{Tr}(\dot{\mathbf{Y}}^\top\mathbf{M}\dot{\mathbf{Y}}) - \mathrm{Tr}(\boldsymbol{\Lambda}\mathbf{M}^\top\boldsymbol{\Lambda}\mathbf{M}) \right\},$$

to obtain

$$\min_{\mathbf{Y} \in \mathbb{R}^{2k \times T}} \min_{\mathbf{W} \in \mathbb{R}^{2k \times d}} \max_{\mathbf{M} \in \mathbb{R}^{2k \times 2k}} L(\mathbf{W}, \mathbf{M}, \mathbf{Y}), \tag{18}$$

where

$$L(\mathbf{W}, \mathbf{M}, \mathbf{Y}) = -\frac{4}{T}\mathrm{Tr}(\dot{\mathbf{Y}}^\top\mathbf{W}\dot{\mathbf{X}}) + 2\,\mathrm{Tr}(\mathbf{W}^\top\mathbf{W}) + \frac{2}{T}\mathrm{Tr}(\dot{\mathbf{Y}}^\top\mathbf{M}\dot{\mathbf{Y}}) - \mathrm{Tr}(\boldsymbol{\Lambda}\mathbf{M}^\top\boldsymbol{\Lambda}\mathbf{M}). \tag{19}$$

After interchanging the order of optimization, we arrive at the objective

$$\min_{\mathbf{W} \in \mathbb{R}^{2k \times d}} \max_{\mathbf{M} \in \mathbb{R}^{2k \times 2k}} \min_{\mathbf{Y} \in \mathbb{R}^{2k \times T}} L(\mathbf{W}, \mathbf{M}, \mathbf{Y}). \tag{20}$$

In the offline setting, we first optimize over $\dot{\mathbf{Y}}$ and find that $\dot{\mathbf{Y}} = \mathbf{M}^{-1}\mathbf{W}\dot{\mathbf{X}}$, or equivalently, $\mathbf{Y} = \mathbf{M}^{-1}\mathbf{W}\mathbf{X}$. We then take a gradient descent-ascent step with respect to $(\mathbf{W}, \mathbf{M})$:

$$\mathbf{W} \leftarrow \mathbf{W} + \eta\left(\frac{1}{T}\dot{\mathbf{Y}}\dot{\mathbf{X}}^\top - \mathbf{W}\right), \qquad \mathbf{M} \leftarrow \mathbf{M} + \eta\left(\frac{1}{T}\dot{\mathbf{Y}}\dot{\mathbf{Y}}^\top - \boldsymbol{\Lambda}\mathbf{M}\boldsymbol{\Lambda}\right). \tag{21}$$

In the network the matrix $\mathbf{W}$ plays the role of feedforward synaptic weights and the matrix $\mathbf{M}$ encodes the recurrent lateral synaptic weights. In the online setting, at each time step $t$, the network receives an input $\mathbf{x}_t$ and computes the output $\mathbf{y}_t$ by running the recurrent neural dynamics to equilibrium:

$$\frac{d\mathbf{y}_t(\gamma)}{d\gamma} = \mathbf{W}\mathbf{x}_t - \mathbf{M}\mathbf{y}_t(\gamma) \qquad \Rightarrow \qquad \mathbf{y}_t = \mathbf{M}^{-1}\mathbf{W}\mathbf{x}_t. \tag{22}$$

The network then computes $\dot{\mathbf{y}}_t := \mathbf{y}_t - \mathbf{y}_{t-1}$ and the synaptic weights are updated according to the local learning rules

$$\mathbf{W} \leftarrow \mathbf{W} + \eta(\dot{\mathbf{y}}_t\dot{\mathbf{x}}_t^\top - \mathbf{W}), \qquad \mathbf{M} \leftarrow \mathbf{M} + \eta(\dot{\mathbf{y}}_t\dot{\mathbf{y}}_t^\top - \boldsymbol{\Lambda}\mathbf{M}\boldsymbol{\Lambda}). \tag{23}$$

## 5.2 Derivation of the second layer

The first layer of the network consists of $2k$ neurons and it transforms the inputs so that the 2D subspaces are represented by pairs of neurons. The second layer of the network consists of $k$ neurons with each neuron computing an angle $\theta_t^i$. The $i^{\text{th}}$ neuron of the second layer receives inputs $(y_{t-1}^{2i-1}, y_t^{2i-1}, y_{t-1}^{2i}, y_t^{2i})$ from the $(2i-1)^{\text{st}}$ and $2i^{\text{th}}$ neurons of the first layer, which satisfy the relation

$$\begin{bmatrix} y_t^{2i-1} \\ y_t^{2i} \end{bmatrix} = \begin{bmatrix} \cos\theta_t^i & -\sin\theta_t^i \\ \sin\theta_t^i & \cos\theta_t^i \end{bmatrix} \begin{bmatrix} y_{t-1}^{2i-1} \\ y_{t-1}^{2i} \end{bmatrix}. \tag{24}$$

Therefore,

$$\theta_t^i = \arctan\left( \frac{y_{t-1}^{2i-1} y_t^{2i} - y_{t-1}^{2i} y_t^{2i-1}}{y_{t-1}^{2i-1} y_t^{2i} + y_{t-1}^{2i} y_t^{2i-1}} \right), \tag{25}$$

which we define to be the output of the $i^{\text{th}}$ neuron in the second layer. For each $i = 1, \ldots, k$, the computation $y_{t-1}^{2i-1} y_t^{2i} - y_{t-1}^{2i} y_t^{2i-1}$ can be implemented in a circuit that resembles the Hassenstein-Reichardt detector, Figure 2(a).

---

**Algorithm 2:** The PCA algorithm

> **initialize** $\mathbf{W}$, $\mathbf{M}$ positive definite, $\mathbf{x}_0 = \mathbf{0}$, $\mathbf{y}_0 = \mathbf{0}$
> **for** $t = 1, 2 \ldots$ **do**
> $\quad \mathbf{y}_t = \mathbf{M}^{-1}\mathbf{W}\mathbf{x}_t$
> $\quad \mathbf{W} \leftarrow \mathbf{W} + \eta((\mathbf{y}_t - \mathbf{y}_{t-1})(\mathbf{x}_t - \mathbf{x}_{t-1})^\top - \mathbf{W})$
> $\quad \mathbf{M} \leftarrow \mathbf{M} + \eta((\mathbf{y}_t - \mathbf{y}_{t-1})(\mathbf{y}_t - \mathbf{y}_{t-1})^\top - \mathbf{\Lambda}\mathbf{M}\mathbf{\Lambda})$
> $\quad$ **for** $i = 1, \ldots, k$ **do**
> $\qquad \theta_t^i = \arctan\left( \frac{y_{t-1}^{2i-1} y_t^{2i} - y_{t-1}^{2i} y_t^{2i-1}}{y_{t-1}^{2i-1} y_t^{2i} + y_{t-1}^{2i} y_t^{2i-1}} \right)$
> $\quad$ **end for**
> **end for**

---

# 6 Simulations

Simulations here are intended to demonstrate the following properties of our algorithms. The first property is their ability to learn arbitrary toroidal groups from data and estimate the speed of transformations. The second property is the ability to generate filters that match theoretical predictions. Finally, we verify that the network output is indeed informative of the transformation by performing reconstruction of the transformed image.

For an arbitrary toroidal group the algorithm must correctly recover 2D subspaces and estimate rotation angles. We created random toroidal groups with three 2D subspaces in 10-dimensional space. An orthogonal matrix $\mathbf{Q}$ defining the group was generated randomly, then rotations angles in each subspace were generated with independent normal distributions with means $0.3, 0.2, 0.1$, and corresponding standard deviations $0.4, 0.3, 0.2$. These rotations were applied to vectors generated from a standard normal distribution in 10 dimensions. In total, $10^6$ samples and rotations were generated and given as input to both algorithms. Learning rates were manually selected to be $5 \cdot 10^{-4}$ for both algorithms. This was repeated 10 times with random initialization. Simulation results are shown in Fig. 4. Subspace fit loss is evaluated as one minus cosine of the angle between the true and estimated planes. Angle estimates are depicted for the last 1000 iterations and pooled randomly from all 10 runs. This experiment took 14 minutes total on a MacBook Pro with 3.5 GHz Dual-Core Intel Core i7 processor.

To study filters, i.e., pairs of vectors for each 2D subspace generated by our algorithms, we experiment with image rotations and translations. As opposed to rotations of abstract vectors above, here 2D subspace angles are correlated. We used natural images from the Van Hateren database [8] and randomly selected $10^6$ patches of size $16 \times 16$ from image frames. The resulting images were then rotated each by random angles around the center using bi-linear approximation and ignoring pixels outside the central circle. The obtained image pairs were used as inputs to SVD algorithm, seeking

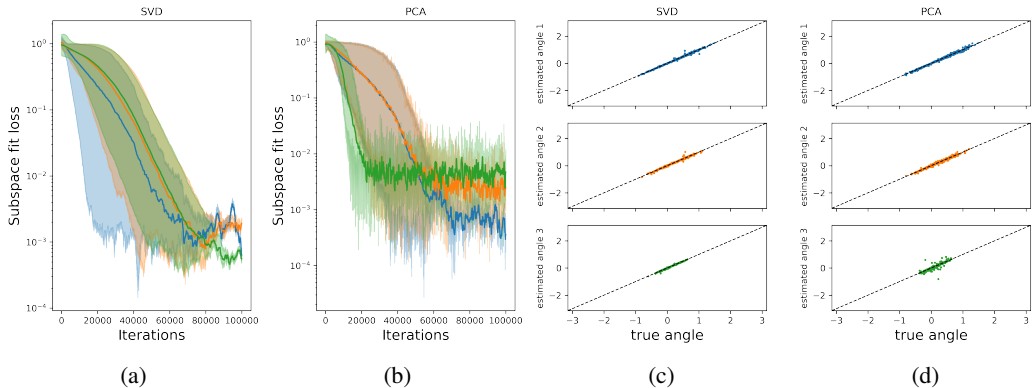

(a)          (b)          (c)          (d)

Figure 4: Learning of toroidal group in vector space. (a) and (b): Convergence of the subspace fit loss versus number of iterations. Blended area shows the range between maxima and minima across all 10 runs; dense line shows mean across runs, median-filtered over 500 iterations. (c) and (d): Angle estimation for each subspace.

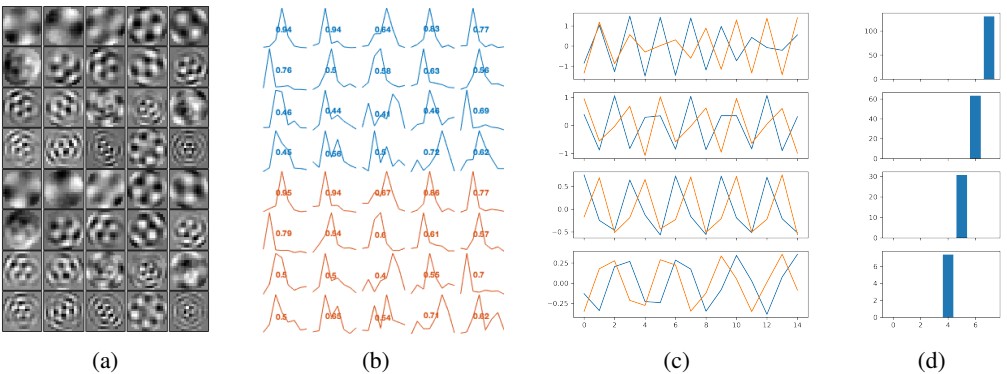

(a)          (b)          (c)          (d)

Figure 5: Filters learned by our algorithms. (a) Filters obtained by the SVD algorithm for rotations of 2D images; left (right) singular vectors are shown in the top (bottom) half. (b) Angular power spectrum for each filter in (a). (c) Filters obtained by the PCA algorithm for translations of 1D images. (d) The power spectrum of each quadrature pair from (c).

20 top pairs of filters. The resulting filter images are presented in Fig. 5(a); they are similar to those obtained from theory in [3], also to those obtained from random images in [5].

Next we provide a quantitative evaluation of the obtained filters versus theoretical predictions. Ideally, in polar coordinates the angular part of Fourier spectrum would have a sharp peak. The estimated angular parts of the spectrum for each filter are shown in Fig. 5(b). Details of this estimation and also additional experiments with rotation of images are presented in Supplement C.

Sub-pixel translations of 1D images can be approximated by cyclic shifts, which are rotations of the image vector. We generated $25 \times 10^5$ 15-pixel 1D images from the standard normal distribution. Top 4 filter pairs obtained by PCA algorithm are shown in Fig. 5(b); they look like Fourier pairs, as expected, and power spectrum of each pair in Fig. 5(c) confirms that. Execution took 5 minutes.

We next demonstrate that the transformations learned by our network on one set of images can be used to transform another set of images. To this end, we use the data from the image rotation experiment above and, using a multi-layer perceptron, fit a function that would predict a transformed image $\mathbf{x}_t$ given the source image $\mathbf{x}_{t-1}$ and the output $\hat{\boldsymbol{\theta}}_t$ of Algorithm 1. This function is then applied to completely different images: digits from the MNIST dataset [9]. We compare digit images transformed this way to directly rotating images with bilinear interpolation of pixel values (the ground truth), Figure 6. The details of this experiment and additional results are presented in Supplement D.

All code for these experiments is included in the Supplementary material.

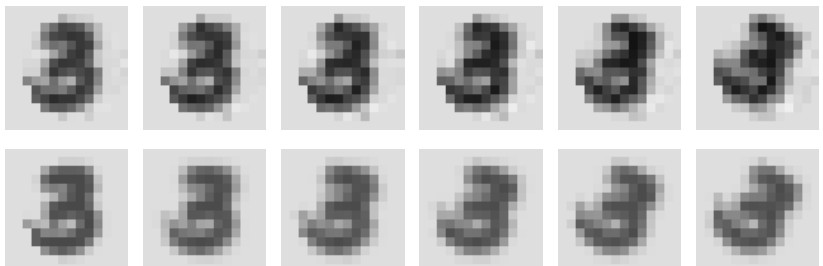

Figure 6: Top row: digit rotations using the transformation learned by the SVD algorithm; Bottom row: digit transformations using image rotation (ground truth).

# 7 Contributions and limitations

The main contribution of this work is the derivation of two biologically plausible online algorithms that can learn toroidal groups of transformations, detect transformations and measure their velocity. By assuming an irreducible representation, as used in [3; 5], we show that a toroidal group can be learned using SVD of the average bivector between vectorized frames before and after transformation, or alternatively using PCA of differences between these vectorized frames. Direct computation of bivector or covariance matrices would involve multiplication of inputs, which is biologically implausible. This is avoided here by representing and learning singular vectors and PCA weight vectors as synaptic weights.

The SVD approach involves modification of a previously known online algorithm for SVD of cross-covariance matrix, and for the case of the top irreducible component we devise a biologically plausible implementation. The approach based on PCA applies a previously developed algorithm for PCA to a new context. The two-layer network that is obtained fully adheres to the principles of biological plausibility outlined in [4]. The ability of algorithms to learn subspaces, measure speed of transformations, generate filters approximating theoretical predictions, and produce output sufficiently informative to generate transformed images, is validated by simulations.

Our work is limited in that we only consider transformations that can be seen as rotations of vectors. This covers important cases of image rotations and sub-pixel shifts but omits several other types. Another limitation comes from the assumption that all transformations observed by our networks come from one commutative group (toroid). We intend to overcome these limitations in our continuing work.

Our work generates experimentally testable hypotheses that can be searched for in nascent connectomics and physiological datasets from various species and brain regions. The relation of our model to actual neurobiological circuits in living organisms is only conjectural and there remain hurdles in turning our algorithms into biologically realistic models. However, our approach can potentially prove beneficial for neuromorphic hardware.

## Acknowledgments

We are grateful to Y. Bahroun and J. Moore for discussion and helpful suggestions, to P. Gunn for editing, and to L. Reading-Ikkanda for help with figures.

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
