# OpenReview forum: "Biological Learning of Irreducible Representations of Commuting Transformations"
_NeurIPS.cc/2022/Conference — NeurIPS 2022 Accept_

### Official Review · Reviewer_pueb · 2022-07-04

**Rating:** 8
**Confidence:** 3
**Soundness:** 4 excellent
**Presentation:** 4 excellent
**Contribution:** 4 excellent

**Summary:**

The authors depict two novel biologically plausible algorithms for online estimation of transformations using irreducible representations. By building on prior work, existing algorithms making use of commutative Lie groups for decomposing transformations are altered for biological plausibility. Possible neural implementations are depicted which can be searched for in connectomics. The results are evaluated experimentally on two small examples.

**Questions:**

How do the non-biologically plausible SVD-based algorithm and the altered version compare in terms of accuracy?

**Limitations:**

Limitations, such as biological implausibility of the original SVD based algorithm, are stated clearly.

**Strengths And Weaknesses:**

Strengths:
- Originality: The general idea of representing Rotations by toroidal groups is not new but nonetheless interesting and coming up with actual biologically plausible implementations is invaluable.
- The text is accessible: starting with a general introduction and good explanation of the problem at hand.
- All Proofs including proof of convergence for "cross-power iteration" are readily available in the appendix


Weaknesses:
- Not clear which parts are contributions, i.e. labeled as "Theorem"
- Sadly no code provided
- Quantitative comparison of the experimental results with "biologically implausible" counterparts (i.e. Cohen and Welling) would've been interesting

Suspected errors:
- Line 73: canonical (ly)
- Line 227: canter

---

> ### Author Response · Authors · 2022-08-02
> **Response to Reviewer pueb**
>
> We thank the reviewer for their feedback and constructive suggestions. We are greatly encouraged that the reviewer finds the proposal of biologically plausible algorithms valuable. We are also pleased that the reviewer finds our exposition accessible. Below we address some of the concerns of the reviewer.
>
> > Not clear which parts are contributions, i.e. labeled as "Theorem"
>
> The main contribution of this work is the derivation of two biologically plausible online algorithms for learning toroidal group transformations. One approach is based on SVD of the bivector between consecutive frames and involves deriving a novel algorithm for SVD. The other approach is based on PCA of the discrete time derivative of the time series and applies previously developed algorithms for PCA to a new context. We will highlight the novel contributions of the work more clearly by adding a short contributions section. We will also clarify this in the text.
>
> > Sadly no code provided.
>
> We thank the reviewer for highlighting this. The omission of the code was unintentional and we have now added the code in the supplementary materials.
>
> > Quantitative comparison of the experimental results with "biologically implausible" counterparts (i.e. Cohen and Welling) would've been interesting
>
> We agree with this. Cohen and Welling (2014) performed similar experiments with rotations of randomly generated images. Unfortunately, no code was provided and no quantitative measures were presented in that work to compare against. We are now adding a quantitative comparison against theoretical predictions, Section 6 and Appendix C.
>
> > How do the non-biologically plausible SVD-based algorithm and the altered version compare in terms of accuracy?
>
> The altered version converges to the correct result, however the speed of convergence is somewhat slower (i.e. is less sample efficient).

---

### Official Review · Reviewer_nuRj · 2022-07-11

**Rating:** 6
**Confidence:** 4
**Soundness:** 2 fair
**Presentation:** 2 fair
**Contribution:** 2 fair

**Summary:**

The present paper studied the neural implementation of learning irreducible representation of commuting group transformations. Specifically, using the 2d rotation group as an example, the paper studied how the neural network learns the rotation groups by implementing different algorithms, including singular value decomposition and principal component analysis, and different algorithms eventually lead to different network architecture.

**Questions:**

- The Introduction (lines 44-45) mentions that an advantage of learning irreducible representation solves the need for multiplicative units and only a linear number of synaptic connections will be needed per neuron. However later in the Result section, I don’t see a detailed explanation of how this problem is solved in the proposed algorithms. Some elaborations are needed.

- Both models (Fig. 2 and Fig. 3) explicitly output the rotation angles $\theta$. In contrast, there are solid neuroscience studies that the angle is represented in a neural population code and can be read out by a linear decoder such as population vector. I am wondering how much the proposed network model and learning rules will be changed if we use a neural population code to represent the rotation angle.


**Ethics Review Area:**

["I don’t know"]

**Limitations:**

- Both models considered in this paper only passively estimate the rotation angle. It would be more interesting to synthesize the transformed images, as in an autoencoder framework.

**Strengths And Weaknesses:**

It is a fundamental question that how neural circuits learn the commutative group transformations. This study provides two novel solutions which rely on single neuron mechanism (implementing SVD) and network mechanism (implementing PCA) respectively to learn the group transformations. The network model implementing PCA (similarity matching specifically) to learn transformations is based on a recent study (ref. 1) so it is not clear how much difference of this part compared with the mechanism presented in ref. 1. The whole paper is structure-wise but some presentation is not clear and the comparison with earlier studies was not done systematically (see details below).

### Writing
- Eq. 3: the index t should be t-1 if I understood correctly.

- Line 141: should the $\sigma$ be the g in Lemma 1? If so, please use consistent notations across the manuscript.

- It is not clear the derivation in Eq. 18 based on Eq. 16 and some motivations are missing. Also, it is not clear the meaning of matrices W and M in Eq. 18 and I need to guess whether they correspond to feedforward and recurrent weight matrices respectively.

- Fig. 5a: not clear which algorithm leads to the presented filters. That is, are those filters learnt from the SVD (representing the u and v), or from the network implementation of PCA (W and M)?

---

> ### Author Response · Authors · 2022-08-02
> **Response to Reviewer nuRj (2/2)**
>
> > Both models considered in this paper only passively estimate the rotation angle. It would be more interesting to synthesize the transformed images, as in an autoencoder framework.
>
> We agree that the synthesis of transformed images is an important and interesting problem. However, this goes beyond the scope of our current work. Our submission can indeed be viewed as a first step in this direction, since estimating the transformation angle is presumably necessary for synthesis. On a biological level, we are not aware of any evidence that a full reconstruction of visual inputs is performed in the brain.

---

> ### Author Response · Authors · 2022-08-02
> **Response to Reviewer nuRj (1/2)**
>
> We thank the reviewer for their detailed feedback and constructive suggestions. Some writing mistakes have been fixed in the text basing on reviewer's comments. Below we address some of the reviewer's concerns.
>
> > The network model implementing PCA (similarity matching specifically) to learn transformations is based on a recent study (ref. 1) so it is not clear how much difference of this part compared with the mechanism presented in ref. 1.
>
> This work is different from that of Ref. 1 in two ways. First, a major limitation in ref. 1 is that they work with infinitesimal transformations whereas we can also handle larger transformations. This allows us to apply our algorithm in a much broader range of settings. Second, in ref. 1, they use the outer product $(x_t-x_{t-1})x_t^\top$ which leads to multiplicative synapses. We instead use bivectors and avoid this requirement. We will clarify these differences in the text.
>
> > It is not clear the derivation in Eq. 18 based on Eq. 16 and some motivations are missing.
>
> We thank the reviewer for pointing out the lack of clarity and typos in our manuscript. Eq. 18 is obtained by substituting in with Legendre transforms for the terms Tr $(\dot X^\top\dot X\dot Y^\top\dot Y)$ and Tr $(\dot Y^\top\Lambda^{-1}\dot Y\dot Y^\top\Lambda^{-1}\dot Y)$. The motivation for these substitutions is to encode the covariance matrices $\frac1T\dot X\dot Y^\top$ and $\frac1T\dot Y\dot Y^\top$ as synaptic weight matrices so that we can obtain an online learning algorithm. We will elaborate on this further in the manuscript.
>
> > Also, it is not clear the meaning of matrices W and M in Eq. 18 and I need to guess whether they correspond to feedforward and recurrent weight matrices respectively.
>
> The matrix $W$ corresponds to feedforward weights and the matrix $M$ corresponds to recurrent lateral weights. We will clarify these further in the manuscript.
>
> > Fig. 5a: not clear which algorithm leads to the presented filters. That is, are those filters learnt from the SVD (representing the u and v), or from the network implementation of PCA (W and M)?
>
> We thank the reviewer for pointing this out. We clarified this in the text.
>
> > The Introduction (lines 44-45) mentions that an advantage of learning irreducible representation solves the need for multiplicative units and only a linear number of synaptic connections will be needed per neuron. However later in the Result section, I don’t see a detailed explanation of how this problem is solved in the proposed algorithms. Some elaborations are needed.
>
> In ref. 1, the authors use the outer product $x_t(x_t-x_{t-1})^\top$, which is vectorized and represented by neural activities (denoted by $\chi_t$). Calculating this outer product requires $d^2$ multiplicative units. The filters are then learned by performing PCA or nonnegative matrix factorization on $\chi_t$ (hence the number of synapses scales as $d^2$ synapses).
>
> In this work, by assuming an irreducible representation, we show that the filters can be learned as the singular vectors of $B=\langle B_t\rangle_t$, where $B_t$ is the bivector $x_tx_{t-1}^\top-x_{t-1}x_t^\top$, or the principal components of the time series $\{x_t-x_{t-1}\}$ (assuming $x_t$ is white).       This allows us to compute the SVD without explicitly computing $B$ in our online algorithm, thus avoiding the need for multiplicative synapses or a quadratic number of neurons. Rather, in our algorithm, each neuron computes projections of the form $u^\top B_tv$, which can be computed using $2d$ synapses (which are encoded in $u$ and $v$). Similarly, the PCA algorithm only requires $d$ (feedforward) synapses per neuron. We will clarify this point further in the manuscript.
>
>
> > Both models (Fig. 2 and Fig. 3) explicitly output the rotation angles $\theta$. In contrast, there are solid neuroscience studies that the angle is represented in a neural population code and can be read out by a linear decoder such as population vector. I am wondering how much the proposed network model and learning rules will be changed if we use a neural population code to represent the rotation angle.
>
> Indeed, angles are often encoded in the brain using a population code, for example in the case of heading direction or wind direction. However in our work, the transformation angle is the amount of rotation of the image vector
> *per unit of time*, so it is better thought of as an angular velocity, or the speed of the transformation. This is often encoded in single neurons in the brain. For instance, the L1 and L2 neurons in the fly lamina encode the magnitude of changes in light intensity. Projection neurons in the invertebrate olfactory system also encode the magnitude of changes in their inputs (from olfactory receptor neurons).

---

> ### Comment · Reviewer_nuRj · 2022-08-09
> **Thanks for the author's reply**
>
> The author's reply clearly states the difference between the current manuscript and the ref. 1. And I am positive that the typo and clarity of the writing can be improved later by the authors.

---

### Official Review · Reviewer_YACa · 2022-07-13

**Rating:** 6
**Confidence:** 3
**Soundness:** 3 good
**Presentation:** 3 good
**Contribution:** 2 fair

**Summary:**

This manuscript proposes biologically plausible algorithms for learning group transformations from sequences of observations. Two algorithms are proposed, one based on SVD and the other on PCA. The performance of these algorithms is evaluated in synthetic experiments.


**Questions:**

The assumption of line 89-90 seems fundamental to be able to the whole paper, but are not discussed. What do they entail?
One particular concern is the independence between $A_t$ and $x_t$, as it is likely that in many practical settings, transformations are performed depending on the previous state.
Moreover, assuming the the pixel intensity vector is white is also very unusual and unlikely in practice.

In addition, the experiment seem to assume that rotations in each subspace are also chosen independently, which is not necessarily the case in practice, and may play a key role in identifying the subspaces.

**Limitations:**

Limitations are not discussed.

**Strengths And Weaknesses:**

Strengths:
- the authors address a very interesting and challenging question: biological plausibility of the learning of transformation representations,

Weaknesses:
- the setting is quite restrictive, reducing the problem to previously proposed biologically plausible algorithms for PCA and SVD.
- experiments are very toy-like, with very simplified assumptions whose influence is not tested,
- relation to previous work on biologically inspired PCA and other such algorithms is elusive. It is difficult to figure out what is the originality of the proposed approaches, and why previous approaches could not directly be applied.

---

> ### Author Response · Authors · 2022-08-02
> **Response to Reviewer YACa**
>
> We thank the reviewer for their feedback and constructive suggestions. We are encouraged that the reviewer finds the problem of finding bio-plausible algorithms of transformation learning interesting and relevant. Below we address some of the concerns of the reviewer.
>
>
> > The setting is restrictive, reducing the problem to previously proposed biologically plausible algorithms for PCA and SVD.
>
> We agree with the reviewer that the setting of the problem in terms of the toroidal group is restrictive. However, we believe that the reduction of this difficult problem to a simpler problem is non-trivial and novel. The use of SVD and PCA in the context of biological learning of finite transformations is new. Furthermore, the proposed biologically plausible algorithm for SVD is itself a novel contribution.
>
> > Experiments are very toy-like, with very simplified assumptions whose influence is not tested
>
> We have expanded our numerical simulations section to include an experiment on natural images and find that our algorithms' performance here is similar to that on the synthetic dataset (see Appendix C for more details). However we agree with the reviewer that the experiments are of small scale. We designed the experiments to prove the viability of our algorithms in principle.
>
> > Relation to previous work on biologically inspired PCA and other such algorithms is elusive. It is difficult to figure out what is the originality of the proposed approaches, and why previous approaches could not directly be applied.
>
> We thank the reviewer for pointing out this shortcoming. We will add further details highlighting the relationship and the novelty. One of our approaches uses previously developed biologically plausible PCA algorithm as a component of motion detection learning. We are not claiming novelty in biologically plausible PCA. However, our use of PCA in the context of motion detection, as well as our biologically plausible SVD, algorithm are novel.
>
> > The assumption of line 89-90 seems fundamental to be able to the whole paper, but are not discussed. What do they entail? One particular concern is the independence between $A_t$ and $x_t$, as it is likely that in many practical settings, transformations are performed depending on the previous state.
>
> This is an important point and we thank the reviewer for bringing it up. It is possible indeed that in many practical settings transformations may depend on the previous state. However, in the context of animal vision, it is critically important for the animal to be able to detect the same kind of transformation being applied to arbitrary image. This need to decouple images and transformations in the study of vision was recognized by Rao and Ruderman in 1999 [1]. Many subsequent works also assume and build on top of this separation [2].
>
> > Moreover, assuming the the pixel intensity vector is white is also very unusual and unlikely in practice.
>
> Whitening of sensory inputs is considered one of the important functions of the sensory systems so we believe it is reasonable to assume that the circuit receives prewhitened inputs. Furthermore, a number of biologically plausible algorithms for input whitening have been proposed and experimentally observed in some cases [3,4].
>
> > The experiment seem to assume that rotations in each subspace are also chosen independently, which is not necessarily the case in practice, and may play a key role in identifying the subspaces.
>
> It is true that in some of our experiments (Fig. 4) subspace rotations are independent. However this is not an assumption used in the proofs. In particular, in our other experiments (Fig. 5) subspace rotations are correlated. We will clarify this in the manuscript.
>
> > Limitations are not discussed.
>
> We will add a section to highlight the shortcomings of the approach. An important limitation of our approach is the assumption that all observed transformations come from one commutative group (toroid). We intend to overcome this limitation in our continuing work.
>
> [1] Rajesh P. N. Rao and Daniel L. Ruderman. Learning lie groups for invariant visual perception. In Advances in neural information processing systems, pages 810—-816, 1999.
>
> [2] Matthias Bethge, Sebastian Gerwinn, and Jakob H. Macke. Unsupervised learning of a steerable basis for invariant image representations. Human Vision and Electronic Imaging XII. Edited by Rogowitz, 6492:64920C–64920C–12, 2007.
>
> [3] Yang Dan, Joseph J. Atick, R. Clay Reid. Efficient Coding of Natural Scenes in the Lateral Geniculate Nucleus: Experimental Test of a Computational Theory, Journal of Neuroscience 15 May 1996, 16 (10) 3351-3362.
>
> [4] Wanner, A.A., Friedrich, R.W. Whitening of odor representations by the wiring diagram of the olfactory bulb. Nat Neurosci 23, 433–442 (2020).

---

> > ### Comment · Reviewer_YACa · 2022-08-08
> > **Thank you. I am still unclear about the actual contribution. It would help to see the rollout**
> >
> > Thank you for your reply. While I agree one should not expect to have the most realistic setting to make an interesting contribution to this question, I am struggling to see concrete and specific evidence that progress is made here.
> >
> > **Novelty of biologically plausible SVD**
> >
> > Regarding your comment:
> > > "We will add further details highlighting the relationship and the novelty. One of our approaches uses previously developed biologically plausible PCA algorithm as a component of motion detection learning. We are not claiming novelty in biologically plausible PCA. However, our use of PCA in the context of motion detection, as well as our biologically plausible SVD, algorithm are novel."
> >
> > I am looking forward to read these further details. In principle, once PCA is solved SVD is solved too, as the singular vectors of matrix A are the eigenvectors of $AA^T$ and $A^TA$ so I am wondering how the need for a new algorithm will be justified.
> >
> > **Whitening assumption and relation to the concrete problem of learning transformations due to movement**
> >
> > For example let us take the question of whitening. As you say:
> > > "Whitening of sensory inputs is considered one of the important functions of the sensory systems so we believe it is reasonable to assume that the circuit receives prewhitened inputs. Furthermore, a number of biologically plausible algorithms for input whitening have been proposed and experimentally observed in some cases [3,4]."
> >
> > The fact that specific neurons may perform some sort of whitening does not mean that this whitening is/can be simply plugged in to a module that extracts an irreducible representation that meaningfully represents *"transformations of stimuli due to their physical movement and changes of perspective"*, state you claim to learn in the introduction. If I take an image of an object undergoing transformations such as translation and rotations, it is very unclear that I would still be able to uncover the structure of those changes after a mapping from the pixel space to the whitened space. Moreover, your setup is framed in the pixel space at the beginning of section 2, not in an abstract space.
> >
> > **Lack of evidence that the structure of image transformations due to movement is learnt**
> >
> > The type of results that you provide on real images in Figure 6b-c also witnesses the mismatch between abstraction of the provided results and the concrete problem that was intended to be addressed. Couldn't you simply leverage the learnt representation to perform rollouts such that we can *see* that a meaningful representation of the *"transformations of stimuli due to their physical movement and changes of perspective"* has actually been learnt? While I would naturally expect something along the lines of Figure 1, all I see is a collection of filters that look similar to the ones obtained with completely synthetic experiments, suggesting some form of generic basis has been learnt, as in many previous works based on biologically plausible algorithm. But unless I missed it, the reader is left with no indication that the concrete and biologically relevant problem intended to be solved has been addressed.

---

> > > ### Author Response · Authors · 2022-08-09
> > > **Response to reviewer YACa's comments (1/2)**
> > >
> > > We thank the reviewer for their thoughtful comments and suggestions.
> > >
> > > ### Novelty of biologically plausible SVD
> > >
> > > > In principle, once PCA is solved SVD is solved too, as the singular vectors of matrix $A$ are the eigenvectors of $AA^\top$ and $A^\top A$ so I am wondering how the need for a new algorithm will be justified.
> > >
> > > This is a good suggestion and we have considered this. While it is true that one can recover the left- and right-singular vectors of the matrix $A$ by performing PCA on $AA^\top$ and $A^\top A$, it is quite challenging to implement this in a biologically plausible neural network (i.e., online with local learning rules).
> > >
> > > Biologically plausible PCA networks have been derived for the special case that the goal is to perform PCA on the covariance of the input dataset (see Oja 1982 and Pehlevan et al. 2015). In our setting, we perform SVD on the matrix $B=\langle x_tx_{t-1}^\top-x_{t-1}x_t^\top\rangle_t$, which is equivalent to performing PCA on $BB^\top$ or $B^\top B$ which are not covariance matrices themselves. We are unaware of any biologically plausible PCA algorithms that can be applied in this setting. In fact, we are unaware on any online algorithm that can solve this problem: it is not clear how to get an online estimate of $BB^\top$ or $B^\top B$ since $\mathbb{E}[B_t]=B$ does not imply that $\mathbb{E}[B_tB_t^\top]=BB^\top$ or $\mathbb{E}[B_t^\top B_t]=B^\top B$. Therefore, our SVD results allow for biologically plausible computations that cannot be performed using existing algorithms.
> > >
> > > ---
> > >
> > > ### Whitening assumption and relation to the concrete problem of learning transformations due to movement
> > >
> > > > The fact that specific neurons may perform some sort of whitening does not mean that this whitening is/can be simply plugged in to a module that extracts an irreducible representation that meaningfully represents "transformations of stimuli due to their physical movement and changes of perspective", state you claim to learn in the introduction. If I take an image of an object undergoing transformations such as translation and rotations, it is very unclear that I would still be able to uncover the structure of those changes after a mapping from the pixel space to the whitened space. Moreover, your setup is framed in the pixel space at the beginning of section 2, not in an abstract space.
> > >
> > > * Whitening of images performed in the brain is traditionally modeled by some kind of low-pass filtering (for study of motion, see e.g. Cadieu and Olshausen 2012). Examples are "Mexican hat" filter or ZCA filtering. These whitening transformations do not map data into some arbitrary space. Rather, the whitened images retain close visual similarity to the original images (see, e.g., Figures 1 and 2 of Pal & Sudeep 2016). Therefore, the transformations of the whitened images are highly informative of the transformations of the original images.
> > >
> > > * Also, consider an unwhitened stream of inputs $\{s_t\}$ such that $s_t=T_ts_{t-1}$, where $T_t$ is a commutative transformation in the sense that $T_t=P\Gamma_t P^{-1}$ for some invertible matrix $P$ and block diagonal matrix $\Gamma_t$ of rotation blocks. Let $\{x_t\}$ be the ZCA whitened transformation of $\{s_t\}$; that is, $x_t=Ws_t$, where $W=C_{ss}^{-1/2}$ and $C_{ss}$ is the covariance of $\{s_t\}$. This whitening transformation can be implemented in a biologically plausible neural network (see Pehlevan and Chklovskii 2015). Then
> > > $$x_t=Ws_t=WT_ts_{t-1}=WT_tW^{-1}Ws_{t-1}=A_tx_{t-1}$$
> > > where $A_t=WT_tW^{-1}$. Suppose $A_t$ lie in a toroidal group; that is, $A_t=Q\Gamma_tQ^\top$ for some fixed $Q$. Then for all $t$,
> > > $$T_t=(Q^\top W)^{-1}\Gamma_t(Q^\top W).$$
> > > Therefore, by finding an irreducible representation of the transformations of the whitened stimuli $A_t$, we also find an irreducible representation of the original transformations. In particular, if one's goal is to reproduce the transformations $T_t$ acting on the pre-whitened signal $\{s_t\}$ (though we do not believe this is the brain's goal), these can be recovered given the *fixed* filters $Q^\top W$ and the *time-varying* angles $\theta_i$.
> > >
> > > * Finally, we note that our approach is more general than looking at pixel space. We can address transformations of any time series. Here we focus on operations on pixel space because they are easy to visualize.

---

> > > > ### Author Response · Authors · 2022-08-09
> > > > **Response to reviewer YACa's comments (2/2)**
> > > >
> > > >
> > > > ### Lack of evidence that the structure of image transformations due to movement is learnt
> > > >
> > > > > The type of results that you provide on real images in Figure 6b-c also witnesses the mismatch between abstraction of the provided results and the concrete problem that was intended to be addressed. Couldn't you **simply leverage the learnt representation to perform rollouts such that we can see that a meaningful representation of the "*transformations of stimuli due to their physical movement and changes of perspective*" has actually been learnt**? While I would naturally expect something along the lines of Figure 1, all I see is a collection of filters that look similar to the ones obtained with completely synthetic experiments, suggesting some form of generic basis has been learnt, as in many previous works based on biologically plausible algorithm. But unless I missed it, the reader is left with no indication that the concrete and biologically relevant problem intended to be solved has been addressed.
> > > >
> > > > * We thank the reviewer for suggesting this idea for validating our approach. Theoretically, we know the transformations $A_t$ can be recovered from the full set of filters $Q$ and all of the angles $\theta_{t,i}$. We now also demonstrate numerically that the output $\hat\theta_{t,i}$ of our network (Algorithm 1) contains sufficient information to reconstruct the transformed images. To this end, we train a network $f(x_{t-1},\hat\theta_{t})$ to minimize the mean-squared error $|f(x_{t-1},\hat\theta_{t})-x_t|^2$ and show that the output of the decoder performs well at reconstructing the transformed images (see Appendix D). In this way, we demonstrate that one can "simply leverage the learnt representation to perform rollouts such that we can see that a meaningful representation of the '*transformations of stimuli due to their physical movement and changes of perspective*' has actually been learnt."

---

### Official Review · Reviewer_vnKM · 2022-07-16

**Rating:** 7
**Confidence:** 4
**Soundness:** 4 excellent
**Presentation:** 4 excellent
**Contribution:** 4 excellent

**Summary:**

The authors present two online biologically plausible algorithms for the irreducible representation learning method proposed by Cohen and Welling (2014).
The first algorithm uses SVD to extract the subspace of maximal average rotation. As part of this algorithm, the authors’ present a novel algorithm for online SVD: cross-power iteration.
The second is based on PCA of time difference vectors and uses local learning rules to preserve biological plausibility.
The authors propose that these models may serve as hypotheses to guide future connectomics and neurophysiology research

Results.
The authors find that both the SVD and PCA methods are able to recover the irreducible representations of SO(2) and S1 on synthetic datasets
These synthetic datasets are sets of random images or vectors sampled from the standard normal distribution, followed by the application of the transformations of interest (translation or rotation).


**Questions:**

see above

**Limitations:**

see above

**Strengths And Weaknesses:**

Strengths:
- The paper contributes two novel and interesting methods for learning the irreducible representations of toroidal groups with biologically plausible mechanisms. This is a valuable contribution for both computational neuroscience (as a hypothesis for neural circuitry) and machine learning (as a novel algorithm for learning and modeling transformation structure from data).

Weaknesses:
- It would be nice to see a quantitative analysis on the learned filters, explicitly comparing to the standard translational and rotational Fourier bases, and reporting how well the learned weights approximate these bases.
- It would be useful to see how well the model trains on natural datasets.

General comments:
- the paper is framed as a neurobiological model, but it really seems more "neuromorphic" as the emphasis is on how to learn all these things with local learning rules in a neural circuit, as opposed to specific neurobiological substrates in the visual system.  There are issues such as how to compute arctan, and many others, that would need to be discussed or elaborated more to think of this in neurobiological terms.
- it will be interesting to see how these results could be extended to real-world applications, such as estimating optical flow from natural videos.

---

> ### Author Response · Authors · 2022-08-02
> **Response to Reviewer vnKM**
>
> We thank the reviewer for their feedback and constructive suggestions. We are encouraged that the reviewer values biologically plausible implementations of algorithms. We are also pleased that the reviewer finds our contribution valuable for both neuroscience and machine learning. Below we address some of the concerns of the reviewer.
>
> > It would be nice to see a quantitative analysis on the learned filters, explicitly comparing to the standard translational and rotational Fourier bases, and reporting how well the learned weights approximate these bases.
>
> We agree with the reviewer that such analysis is important, and we are adding it to the manuscript (Section 6 for  translations, Appendix C for rotations). To summarize, we find that our filters are qualitatively similar to the Fourier modes but are not exactly the same.
>
> > It would be useful to see how well the model trains on natural datasets.
>
> We thank the reviewer for this suggestion. We carried out a numerical experiment by looking at rotations on naturalistic images from the Van Hateren database. We again see that the filters that our algorithm finds are qualitatively similar to Fourier modes (see Appendix C).
>
> > The paper is framed as a neurobiological model, but it really seems more "neuromorphic" as the emphasis is on how to learn all these things with local learning rules in a neural circuit, as opposed to specific neurobiological substrates in the visual system. There are issues such as how to compute arctan, and many others, that would need to be discussed or elaborated more to think of this in neurobiological terms.
>
> We thank the reviewer for this suggestion. We agree that our approach can potentially prove beneficial for neuromorphic hardware and will mention this as one of the implications of our work. We will also clarify that, as the reviewer correctly points out, the relation of our model to actual neurobiological circuits in living organisms is only conjectural and there remain hurdles in turning our algorithm into a biologically realistic model.
>
> > It will be interesting to see how these results could be extended to real-world applications, such as estimating optical flow from natural videos.
>
> This is a great suggestion. We believe that our framework can potentially handle real-world use cases when augmented and properly layered. We thank the reviewer for their encouragement and leave this direction to future work.

---

### Author Response · Authors · 2022-08-02
**General response**

We thank the reviewers for their thoughtful feedback. We are pleased  that they found the subject of our study important and interesting. We are also encouraged that the reviewers believe that the algorithms can be of interest to the ML community and that the novel biologically plausible algorithms would be of great value to the neuroscience community.

We found the reviewer comments very constructive, and in accordance with their suggestions, have made modifications. We summarize the changes and additions here and provide more detailed responses to each reviewer individually.

* **Numerical experiments on naturalistic images.** To verify our algorithm in a more challenging setting, we have performed an experiment looking at transformations on naturalistic images.

* **Quantitative comparison with Fourier basis.** We have included quantitative comparison of the filters obtained in our algorithm (for both naturalistic and synthetic experiment) with Fourier basis filter.

---

### Meta-Review · Area_Chair_qor4 · 2022-08-26

**Recommendation:** Accept
**Confidence:** Certain

**Metareview:**

This manuscript presents novel biologically plausible algorithms for learning representations for Lie groups. The derivation of the algorithms and the networks are based on previously studied biologically plausible networks. Although there are some limitations, the reviewers agree that this work is sound, clearly presented, and represents a valuable contribution to both computational/theoretical neuroscience and machine learning.

**Award:**

No

---

### Decision · Program_Chairs · 2022-09-14

Accept